# Impaired CENP-E Function Renders Large Chromosomes More Vulnerable to Congression Failure

**DOI:** 10.3390/biom9020044

**Published:** 2019-01-26

**Authors:** Laura Tovini, Sarah E. McClelland

**Affiliations:** Barts Cancer Institute, Queen Mary University of London, London EC1M 6BQ, UK; l.tovini@qmul.ac.uk

**Keywords:** CENP-E, chromosome congression, aneuploidy, chromosome identity

## Abstract

It has recently emerged that human chromosomes vary between one another in terms of features that impact their behaviour during impaired chromosome segregation, leading to non-random aneuploidy in the daughter cell population. During the process of chromosome congression to the metaphase plate, chromosome movement is guided by kinesin-like proteins, among which centromere-associated protein E (CENP-E) is important to transport chromosomes along the microtubules of the mitotic spindle. It is known that the inhibition of CENP-E notably impairs alignment for a subset of chromosomes, particularly those positioned close to the centrosome at nuclear envelope breakdown (‘polar chromosomes’); it is, however, not clear whether chromosome identity could influence this process. Since a popular strategy to model aneuploidy is to induce congression defects (for example combining CENP-E inhibitors with mitotic checkpoint abrogation), variance in congression efficiency between chromosomes might influence the landscape of aneuploidy and subsequent cell fates. By combining immunofluorescence, live cell imaging and fluorescence in situ hybridisation, we investigated the behaviour of polar chromosomes and their dependency upon CENP-E-mediated congression in human cells. We observed a bias in congression efficiency related to chromosome size, with larger chromosomes more sensitive to CENP-E inhibition. This bias is likely due to two contributing factors; an initial propensity of larger chromosomes to be peripheral and thus rely more upon CENP-E function to migrate to the metaphase plate, and additionally a bias between specific chromosomes’ ability to congress from a polar state. These findings may help to explain the persistence of a subset of chromosomes at the centrosome following CENP-E disruption, and also have implications for the spectrum of aneuploidy generated following treatments to manipulate CENP-E function.

## 1. Introduction

The ability to maintain chromosome segregation fidelity is a major feature of mitosis. During prometaphase, chromosomes congress towards the spindle equator [1]. Although in mammalian cells chromosomes can congress before becoming bioriented [2], kinesin-like proteins precisely guide chromosome movement during the formation of the metaphase plate. Among them, centromere-associated protein E (CENP-E) facilitates chromosome alignment by assisting their motion towards plus ends of microtubule bundles of the mitotic spindle [2,3]. Moreover, CENP-E is involved, together with other kinesins, in promoting end-on conversion of chromosome-microtubule attachment [4]. Studies characterising the effects of impaired CENP-E function have shown efficient alignment for most chromosomes, but with a subset remaining close to the poles [5,6], suggesting a differential dependency on CENP-E between chromosomes [7]. One cause of this variance is that chromosomes positioned close to the centrosome or outside the interpolar axis at nuclear envelope breakdown are particularly reliant on CENP-E function [5]. We were interested in discovering whether chromosome identity could also influence these dependencies, given recent observations that mammalian chromosomes non-randomly mis-segregate when the mitotic spindle is perturbed [8,9]. Errors in the process of chromosome segregation can lead to aneuploidy, a key hallmark of cancer [10,11,12]. Indeed, CENP-E heterozygous mice have been shown to display increased aneuploidy and tumour formation [13]. Moreover, common strategies to study aneuploidy in cellular models of cancer also employ alteration of CENP-E function. For example, recent strategies to elevate aneuploidy in diploid cells have used CENP-E inhibition coupled to inactivation of the mitotic checkpoint using Mps1 inhibitors [14,15]. Given the open questions about the non-uniform behaviour of chromosomes during impaired chromosome congression, and the unknown impact of CENP-E perturbation on specific human chromosomes, we carefully explored the behaviour of uncongressed chromosomes when CENP-E function is compromised, using live cell imaging and immunofluorescence. We show that polar chromosomes comprise two groups, namely, that some chromosomes are subsequently able to congress while others are terminally uncongressed. Differential ability to congress between these groups is not dependent on microtubule attachment status, position relative to the centrosome, or time spent in mitosis. Furthermore, to detect bias between chromosomes we applied fluorescence in situ hybridisation (FISH) to quantify congression behaviour of each human chromosome. Interestingly, chromosome identity influences the likelihood of remaining perpetually uncongressed following CENP-E inhibition, with large chromosomes more likely to remain perpetually uncongressed.

## 2. Materials and Methods

### 2.1. Cell Culture and Fluorescent Protein Expression

Immortalised retinal pigment epithelium-hTert (RPE1) cells were from ATCC (UK). Stable expression of H2B-RFP in hTERT-RPE1 cells was achieved by transfection with lentiviral construct H2B-RFP (LV-RFP was a gift from Elaine Fuchs; Addgene plasmid # 26001; http://n2t.net/addgene:26001; RRID:Addgene_26001). Cells were grown in DMEM Nutrient Mixture F12 Ham (Sigma-Aldrich Company Ltd., Gillingham, UK) supplemented with 10% FBS (ThermoFisher Scientific, UK) and 100 U Penicillin/Streptomycin (Sigma Aldrich) at 37 °C and 5% CO_2_. These cells also carried a variable penetrance of trisomy 12, as previously observed [8]. Routine test results from mycoplasma were negative and RPE1 cells were subjected to STR (short tandem repeats) profiling to verify their identity using the cell line authentication service from Public Health England in November 2017.

### 2.2. Drug Treatment

Motor activity of CENP-E was inhibited by GSK923295 (Cayman Chemical, Michigan, USA) dissolved in growth medium and used at a final concentration of 20 nM. AuroraB was inhibited by ZM447439 (Cambridge Bioscience, Cambridge, UK) dissolved in DMSO and used at a final concentration of 1 µM. Mitotic centromere-associated kinesin (MCAK) was upregulated by UMK57 (a kind gift from Benjamin Kwok and Duane Compton [16] (the full characterisation of UMK57 will be published elsewhere)).

### 2.3. Immunofluorescence

Cells grown on coverslips were fixed in freshly-prepared PTEMF (0.2% Triton X-100, 0.02 M PIPES (pH 6.8), 0.01 M EGTA, 1 mM MgCl_2_, 4% formaldehyde) for 10 min at room temperature. For cold treatment assays cells were placed on ice for 10 min before fixation. After blocking with 3% BSA, cells were incubated with primary antibodies: α-tubulin at 1:600 (Abcam abID#6046, Cambridge, UK), Centrin3 at 1:500 (Abcam abID#54531), CREST at 1:400 (Antibodies Incorporated, #15-234-0001, Davis, CA), CENP-A at 1:400 (Abcam abID#13939), BubR1 at 1:500 (Cambridge Bioscience), Mad2 at 1:500 (Bethyl Lab, A300–300A). Secondary antibodies used were goat anti-mouse AlexaFluor488 (A11017, Invitrogen, UK), goat anti-rabbit AF594, AF488 (A11012, A11008, Invitrogen), and goat anti-human AF647 (109-606-088-JIR, Stratechor A21445, Invitrogen). DNA was stained for 6 min with DAPI (Roche, UK) and coverslips mounted in Vectashield (Vector H-1000, Vector Laboratories, Peterborough, UK).

### 2.4. Fluorescence In Situ Hybridisation

Cells were grown on glass slides or on coverslips, fixed in cold methanol/acetic acid (3:1), then put through an ethanol dehydration series and air dried. Cells and specific centromere probe (CEP), subtelomere specific probes or breakapart probes (Cytocell, UK) were denatured for 2 min at 75 °C then incubated overnight at 37 °C. The following day, slides were washed with 0.25X SSC at 72 °C followed by a brief wash in 2X SSC, 0.05% Tween. DNA was stained for 6 min with DAPI (Roche) and coverslips mounted in Vectashield (Vector H-1000, Vector Laboratories).

### 2.5. Microscopy

Images were acquired using an Olympus DeltaVision RT microscope (Applied Precision, LLC, USA) equipped with a Coolsnap HQ camera. Three-dimensional image stacks were acquired in 0.2 µm steps, using Olympus 100X (1.4 numerical aperture) or 60X UPlanSApo oil immersion objectives. Deconvolution of image stacks was performed with SoftWorxExplorer (Applied Precision, LLC). H2B-RFP-labelled RPE1 cells were live imaged in four well imaging dishes (Greiner Bio-one, UK). The 20-µm z-stacks (10 images) were acquired using an Olympus 40X 1.3 numerical aperture UPlanSApo oil immersion objective every 3 min for 8 h using a DeltaVision microscope in a temperature and CO_2_-controlled chamber. Analysis was performed using Softworx Explorer.

### 2.6. Preparation of Illustration

Contrast and brightness of the final images were linearly adjusted in Photoshop (Adobe Photoshop CC 2018, USA) and the figures assembled in Illustrator (Adobe Illustrator CC 2018, USA). Graphs were prepared in Prism 5.4 (GraphPad Software, San Diego, CA) and imported in Illustrator.

## 3. Results

### 3.1. A Subset of Chromosomes Remains Perpetually Uncongressed after CENP-E Inhibition

We examined chromosome congression defects in RPE1 cells following small molecule inhibition of CENP-E with GSK923295 [17]. Treatment of cells for 5 h with 20 nM centromere-associated protein E inhibitor (CENP-Ei) resulted in a high proportion of cells with a clear metaphase plate, but with several uncongressed chromosomes in the vicinity of the centrosomes (Figure 1a,b) as previously observed using inhibition or depletion of CENP-E [2,4,6]. To gain more information about the behaviour of chromosome congression under impaired CENP-E function we performed live cell imaging of RPE1 cells stably expressing H2B-RFP (Figure 1c). As expected, the control cells efficiently congressed all chromosomes in 13 ± 3 min following nuclear envelope breakdown (NEBD) and underwent anaphase in 26 ± 3 min (Figure 1d). By contrast, cells treated with CENP-E inhibitor significantly delayed congression (Figure 1c) and thus delayed or failed anaphase onset (Figure 1d). Cells treated with CENP-E inhibitor displayed a distinctive series of chromosome congression behaviours (Figure 1e): Phase I; a rapid initial congression phase that produced a recognisable metaphase plate with in the first 38 ± 12 min of NEBD. Phase II; a subsequent slower, but progressive phase of congression at the rate of ~1 chromosome per hour that lasted until 240 min post-NEBD. Phase III; a paused phase, where the majority of cells that had not yet congressed all chromosomes and entered anaphase remained arrested in prometaphase with 2.56 ± 0.8 chromosomes for the remainder of the movie. For a small subset of cells that could be followed for longer than 300 min post NEBD, we observed the onset of cohesion fatigue, as previously observed under conditions of prolonged metaphase arrest [8,18], where chromosomes were seen scattering from the metaphase plate (Figure 1c) (see Figure A1 for individual cell congression profiles).

### 3.2. Perpetually Polar Chromosomes Are Not Shielded by the Centrosomes

The rapid formation of the pseudo-metaphase plate even when CENP-E function is impaired suggests that many chromosomes are able to rapidly bi-orient in the absence of lateral microtubule motion, potentially because they are already positioned near the equator of the cell at the moment of nuclear envelope breakdown. However, some chromosomes fail to align during this rapid congression phase and remain at the poles (‘polar chromosomes’ [6]). It is known that initial position at NEBD influences the propensity of chromosomes to fail initial congression and become polar [5]. However, we were interested in understanding why a subset of these polar chromosomes were then able to congress-albeit at a reduced rate-during the progressive phase (phase II, Figure 1e), while others were left stranded at the centrosomes indefinitely (‘perpetually polar chromosomes’). First, we considered the possibility that persistent polar chromosomes were simply positioned behind the centrosomes, preventing congression to the metaphase plate. To test this, we marked centrosomes using anti-centrin 3 antibodies, and compared the distribution of chromosome positions relative to the centrosomes (‘inner’ or ‘outer’ (Figure 2a,b)) at two timepoints following CENP-E inhibition. Since cells continually enter mitosis during CENP-Ei treatment, we sub-categorised cells based on the number of uncongressed chromosomes to more accurately assess chromosome position relative to time spent in CENP-Ei (4–6 uncongressed chromosomes (early cells) or 1–3 uncongressed chromosomes (late cells); Figure 2c). We reasoned that if a subset of chromosomes was shielded behind the pole, that these would be refractory to early congression, and there should be a higher proportion of outer (shielded) chromosomes with time. There was a slight increase in the percentage of outer chromosomes in later (1–3 uncongressed chromosomes) cells at the 60 min timepoint. However, across all cells and both timepoints, the majority of chromosomes were positioned inside or parallel to the centrioles (Figure 2d) suggesting that centrosome shielding was unlikely to fully explain the failure of perpetually arrested chromosomes to congress.

### 3.3. Kinetochore-Microtubule Dynamics Do Not Influence the Behaviour of Polar Chromosomes

The inhibition of CENP-E is proposed to impair the conversion from lateral to end-on kinetochore-microtubule attachment [4] and therefore uncongressed chromosomes are known to be laterally attached [5,19,20]. However, we wondered if a subset of the perpetually polar chromosomes might have formed syntelic attachments (both kinetochores (KTs) attached to microtubules (MTs) emanating from the same centrosome) that could impair their congression. Indeed, polar chromosomes frequently appeared positioned with both kinetochores oriented toward the centrosome (see Figure 2a). To investigate whether syntelic attachments were present at polar chromosomes, we performed a brief cold treatment to remove unstable spindle microtubules and allow observation of stable kinetochore microtubule (KT-MT) attachments using tubulin and CENP-A. In contrast to the clear end-on (and amphitelic) attachments observed at chromosomes within the metaphase plate (Figure 3b, left panel), we never observed any end-on attachments at polar chromosomes, suggesting a lack of syntelic or monotelic attachment. Additionally, we could often see polar chromosomes perpendicular to MTs (Figure 3b, right panel). Next, we examined the mitotic checkpoint status of polar chromosomes using antibodies to spindle assembly checkpoint proteins Mad2 and BubR1 which revealed that the vast majority of KTs efficiently loaded Mad2 and BubR1 in agreement with a lack of end-on MT attachment (Figure 3c,d). These analyses suggested that polar chromosomes were likely to be associated with the microtubule lattice. To systematically rule out the presence of syntelic attachment that may have evaded detection using imaging approaches we functionally modulated KT-MT dynamics during the progressive congression phase. We reasoned that increasing KT-MT turnover would promote release of syntelic attachments and rescue polar chromosomes, whilst decreasing turnover would further impair polar chromosome congression. To do this, we first treated cells with CENP-E inhibitor for 1 h, to allow cells to enter mitosis and establish a metaphase plate with a few uncongressed chromosomes. Then we added either a potentiator of MCAK MT depolymerase (UMK57 [16]) or an Aurora B inhibitor (ZM447439) to increase or decrease KT-MT turnover respectively during the final hour of CENP-E inhibition (Figure 3e). Alone, UMK57 and ZM447439 treatments each resulted in failure to efficiently form metaphase plates, confirming their ability to deregulate KT-MT dynamics as expected (Figure 3f,g). However, neither UMK57 nor ZM447439 treatment changed the number of polar chromosomes remaining after 2 h CENP-E inhibition (Figure 3h,i). Taken together, these data suggest that the failure of polar chromosomes to congress is not due to rare syntelic or other end-on MT attachments to the centrosomes.

### 3.4. Congression of Large Chromosomes Is Particularly Impaired by CENP-E inhibition

We next wondered if a systemic cellular process might be limiting the time available for progressive chromosome congression. We rarely observed any congression events after 240 min post-NEBD (see Figure 1e), suggesting that extended time in mitosis might ultimately lead to a cellular state that precluded further chromosome congression. However, live cell analyses showed that a proportion (56%) of CENP-Ei-treated cells did proceed to anaphase during this time (see Figure 1d). In accordance with the efficient loading of mitotic checkpoint proteins Mad2 and BubR1 at polar chromosomes (see Figure 3c,d), we noted from live cell imaging that cells correctly aligned all chromosomes before undergoing anaphase (see Figure 1) at least within the time resolution of the time-lapse filming (3 min). To examine why some cells were able to congress all chromosomes and progress into anaphase whereas some cells remained arrested in prometaphase, we compared parameters between cells that did, or did not ultimately enter anaphase. There was no difference in the number of uncongressed chromosomes at the start of the progressive congression phase (Phase II; Figure 4a). Both classes of cells also proceeded in a similarly progressive and consistent manner through this congression phase though we noted a small difference in the rate of congression (Figure 4b,c). These similarities suggested that the factor influencing whether cells could congress all their chromosomes and proceed to anaphase was not a systemic state related to the time cells had spent in prometaphase, and was not a function of the initial number of polar chromosomes. Therefore, we wondered instead whether the identity of the polar chromosomes might dictate whether they could be congressed efficiently in the absence of CENP-E function. To test whether chromosome identity could influence the likelihood of becoming terminally uncongressed, we treated cells with CENP-Ei for 2 h then performed FISH with chromosome-specific probes against all chromosomes. Where specific centromere probes were not available, we used sub-telomere-specific, or interstitial probes (see Table 1 for details). We scored the frequency with which each chromosome was uncongressed in cells where 3 or fewer chromosomes were uncongressed, to ensure we were analysing cells in the ‘paused’ phase (Phase III) of congression (Figure 4d). This analysis revealed a clear bias towards large chromosomes remaining uncongressed; of the twelve largest chromosomes (chromosomes 1–11 and the X chromosome) eight remained uncongressed at higher than expected rates (expected: 4.3% per chromosome), whereas eight of the eleven smaller chromosomes (chromosomes 12–22) were affected less than expected rates (Figure 4e,f). We also noted interesting deviations from this pattern. Chromosomes 5 and 9 were much less frequently affected than chromosomes of similar size. Notably, chromosome 18 was more affected than expected, especially considering its small size.

Cells with 1–3 uncongressed chromosomes analysed in these FISH experiments are likely a mix of cells destined to enter anaphase, and those destined to arrest with perpetually polar chromosomes. Thus, the bias towards larger chromosomes remaining uncongressed likely reflects a combination of two distinct variables; (i) the propensity for a given chromosome to be polar initially, and (ii) the ability of specific chromosomes to congress from a polar starting position. To test the contribution of these variables we analysed the uncongression bias in cells with 4–6 uncongressed chromosomes (‘early’ cells) and compared this to data collected from cells with 1–3 uncongressed chromosomes (‘late’ cells (Figure 4f)). We specifically asked; for chromosomes that are uncongressed above a rate of 4.3% in late cells (e.g.,1, 2, 4, 6, 8 and 18) what is the uncongression bias in ‘early’ versus ‘late’ cells? Interestingly we see a clear general trend (although most comparisons do not reach statistical significance); bias is lower in early cells, suggesting that initial positioning is not the only factor, and that these chromosomes are additionally less able to congress from a polar position. Taken together these data suggest that large chromosomes appear to be more likely to (i) become polar initially after NEBD and also (ii) fail to congress from a polar position.

## 4. Discussion

To date, it has been shown that CENP-E inhibition impairs chromosome congression, with a characteristic subset of chromosomes that fail to initially congress, often as a result of being located outside the interpolar microtubules at NEBD [5]. Here, we used live cell imaging to show that of the initial polar chromosomes, there is a final set that fails to congress during prolonged prometaphase. We show these perpetually polar chromosomes are generally not obviously shielded by centrosomes, and are not attached to centrosomes in a syntelic manner. Analysis of chromosome identity revealed a bias towards larger chromosomes becoming arrested at the centrosome, which appears to be contributed to by both an initial propensity to become polar, coupled to a reduction in congression efficiency from a polar state (Figure 4g). Chromosomes have been shown to exhibit non-random nuclear distribution, with larger chromosomes tending to be at the periphery [21,22]. It is, therefore, a strong possibility that the size bias may reflect the initial arrangement of chromosomes in the nucleus; small chromosomes may be more likely to be positioned close to the site of the metaphase plate and be more likely to encounter MTs at the correct geometry to form end-on attachments even in the absence of CENP-Ei. In contrast, large chromosomes positioned at the periphery of the nucleus would be more likely to become polar since the initial position relative to the centrosome is important in dictating the dependency upon CENP-E for congression [9]. However, there is also evidence that chromosomes are near randomly distributed at the onset of mitosis [23]. This suggests that the position of chromosomes in the nucleus could be relevant for their increased dependency on CENP-E, but that other factors may also be involved. Chromosomes can be excluded from central parts of the spindle by arm ejection forces [5] which would be expected to affect large chromosomes more, therefore this could be another contributing factor to the propensity of large chromosomes to become polar at NEBD. Additionally, one notable class of chromosomes, the acrocentric chromosomes, that contain the ribosomal DNA and are usually positioned close to nucleoli [24] are all in the ‘small’ chromosome category. It is possible that their tethering to the nucleolus renders them more likely to be positioned close to the interior of the nucleus and depend less on CENP-E for metaphase alignment.

Since some, but not all, chromosomes congress during the progressive phase, and since some cells are able to congress all chromosomes, whereas others are not, an additional bias towards large chromosomes may be introduced during this phase. Indeed, the fact that cells that do, or do not, ultimately congress all chromosomes and enter anaphase start with an identical average number of polar chromosomes suggested a bias in the ability of some of these chromosomes to congress. By contrast, if all chromosomes were equally able to congress, we would expect a correlation between a low number of polar chromosomes, and the ability to enter anaphase (which we do not see). This hypothesis is supported by FISH experiments comparing the bias in early and late cells that suggests at least some bias is introduced independently of initial positioning near the centrosome. What might explain the variance in the ability of specific chromosomes to congress from a polar position? Kinetochore-bound dynein provides a force directing uncongressed chromosomes towards microtubule minus ends [25,26], explaining the proximity of uncongressed chromosomes to the centrosome in the absence of end on attachment. Chromokinesins provide the counterbalance force (the polar ejection force) [27], and the balance between these forces could be important to determine whether chromosomes are able to congress in the absence of CENP-E. Interestingly, although large chromosomes should potentially benefit from a higher polar ejection force, the observed enrichment of these chromosomes at centrosomes would support the previous observation that the force balance is in fact tipped towards dynein winning [5]. It has been estimated that centromere strength during end-on attachment should not need to compensate for chromosome size since spindle forces are magnitudes higher than drag produced by chromosomes [28,29]. However, the forces involved in dynein-mediated movements along MTs might be closer to the scale where chromosome size-related viscous drag could be a factor. Large chromosomes may inherently load more dynein to compensate for their size. Under normal conditions this is counterbalanced by CENP-E function, but following CENP-E inhibition large chromosomes are vulnerable to congression failure. Interestingly, recent work has shown that chromosomes with larger centromeres depend less on CENP-E for congression in the Indian Muntjak system [9]. This would suggest that even if large chromosomes do load more dynein, that this might not translate directly into a ‘larger’ kinetochore per se (in terms of providing additional MT binding sites).

There may be clues from the exceptions to the rule of large chromosomes remaining uncongressed. What is it about chromosomes 5, 7 and 9 that mean they escape frequent congression failure? A cursory analysis of basic chromosome biology does not reveal any obvious features of these chromosomes. It could be that these chromosomes inhabit internal chromosome territories despite their large size. Alternatively, there may be important functional differences between human chromosomes that are currently unknown. In the other direction, why is chromosome 18 more vulnerable to congression failure than other small chromosomes? Notably, chromosome 18 is thought to have the longest centromeric region of all the human chromosome (5.4 MB); therefore, it is possible that it overloads dynein relative to chromosome size, resulting in less efficient congression. However, chromosome 17 also has a large centromere (4.1 MB) but is rarely affected. Therefore, there does not appear to be a clear correlation between centromere length as estimated from centromeric array lengths and propensity to remain polar. However, it is not currently possible to make robust conclusions regarding contribution to congression bias from differences in centromere length or size, since these parameters are not currently well defined. Future studies will be required to functionally test centromeric differences that could contribute to variance in congression ability.

It is noteworthy that chromosomes 1, 2 and 18 have previously been found to be more prone to improper segregation during perturbation of mitosis using nocodazole washout [8]. It could be that these chromosomes are vulnerable to defects in both chromosome congression and segregation, although likely via distinct mechanisms. Taken together, a picture is now emerging of how individual chromosome characteristics can influence multiple processes during mitosis [8,9]. Since defects in mitosis can lead to aneuploidy, understanding the rates at which individual chromosomes can become aneuploid under specific impairments of cell division may be important for interpreting aneuploidy landscapes in cancer. In this regard, it is interesting that chromosome congression defects have been observed in colorectal cancer [30] and ovarian cancer (our unpublished observations), although not via deregulation of CENP-E function as far as we are aware.

## 5. Conclusions

In summary, by examining the behaviour of chromosomes most affected by CENP-E inhibition, we found that large chromosomes are particularly vulnerable to congression failure. This is likely to be due to a combination of factors, potentially including nuclear chromosome territories, in addition to other, as yet unknown, features of chromosomes that impact the ability to congress from a polar position. Further investigation of the specific features that render particular chromosomes more dependent on CENP-E will likely provide further insights into fundamental mechanisms controlling movement and accurate segregation of chromosomes during cell division.

## Figures and Tables

**Figure 1 biomolecules-09-00044-f001:**
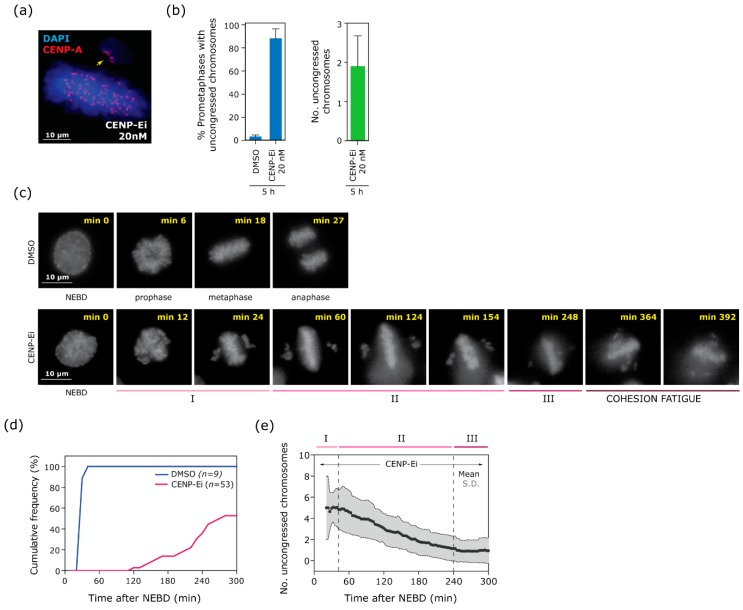
Inhibition of centromere-associated protein E (CENP-E) results in faulty metaphase plate formation and leads to a multi-phase chromosome congression pattern followed by cohesion fatigue. (**a**) Example of immunofluorescence image of retinal pigment epithelium-hTert (RPE1) cells treated for 5 h with centromere-associated protein E inhibitor (CENP-Ei), stained with CENP-A antibody to mark centromeres (red). (**b**) Congression error rate and average number of uncongressed chromosomes per erroneous metaphase. Mean and standard deviation (S.D.) from three independent experiments is shown. Total number of cells analysed: 138 for DMSO treatment, 236 for CENP-Ei treated. Total number of uncongressed chromosomes: 507. (**c**–**e**) Live-cell imaging of RPE-1 cells stably expressing H2B-RFP. Movies started immediately after drug addition, cells were imaged every 3 min for 8 h in total. (**c**) Representative frames of live cell imaging of RPE-1 cells stably expressing H2B-RFP. Top panels: Control (DMSO) cell going into a normal mitosis. Bottom panels: CENP-Ei treated cell undergoing faulty chromosome congression. (**d**) Cumulative frequency from nuclear envelope breakdown (NEBD) to anaphase onset of cells treated with CENP-Ei (red line), compared to control (DMSO) cells (blue line). Timing has been normalised to NEBD (min 0). (**e**) Mean (black dots) and S.D. (grey shade) of uncongressed chromosome number over time in cells treated with CENP-Ei. The top coloured lines reflect three different phases of congression: I) metaphase plate formation, II) progressive chromosome congression, III) pause phase. See also Figure A1 for individual cell plots. Data in (**d**) and (**e**) is from cells imaged for ≥300 min post-NEBD, or undergoing anaphase within 300 min, from three independent experiments (15 cells arrested in prometaphase and 21 cells going into anaphase in total).

**Figure 2 biomolecules-09-00044-f002:**
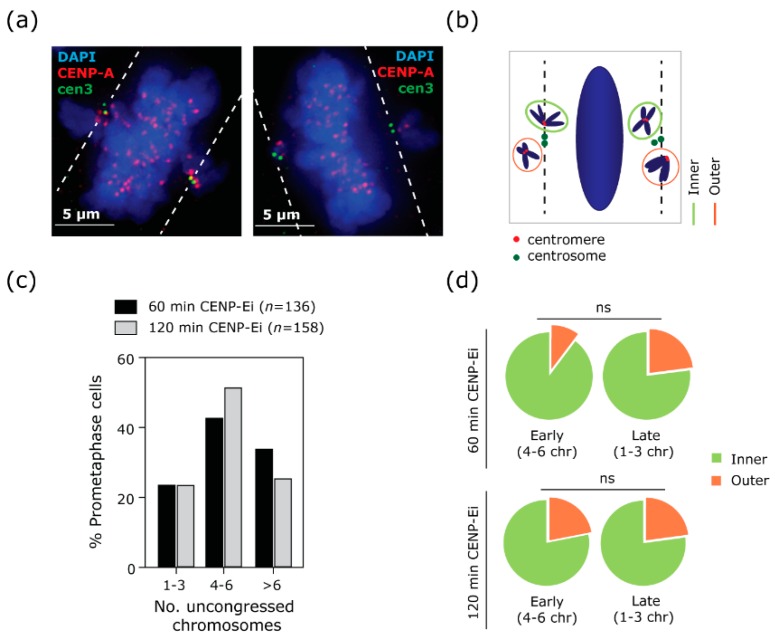
The position of uncongressed chromosomes (chr) does not fully explain their persistence near the centrosomes. (**a**) Example immunofluorescence image of RPE1 cells treated for 2 h with CENP-Ei, stained with CENP-A antibody to mark centromeres (red) and with Centrin3 (cen3) antibody to mark centrosomes (green). In the example, two prometaphases are reported: the left panel shows a prometaphase cell in an early stage (4–6 uncongressed chromosomes), the right panel shows a prometaphase cell in a later stage (1–3 uncongressed chromosomes). A white dashed line, parallel to the metaphase plate, has been used as a reference to categorise the uncongressed chromosomes (also schematised in (**b**)). (**b**) Any centromeric signal that is touching the line or is located between the line and the metaphase plate is classed as “inner”; any centromeric signal which is not touching the line or is distal from the cen3 signal is classed as “outer”. (**c**) Distribution of prometaphase cells with 1–3, 4–6 or more than 6 uncongressed chromosomes at each time point (60 min and 120 min) (*n* = number of cells analysed in each condition). (**d**) Quantification of position of uncongressed chromosomes in early or late phase cells at 60 or 120 min of CENP-Ei treatment, two experiments (125 cells and 676 chromosomes analysed for 60 min treatment; 126 cells and 717 chromosomes analysed for 120 min treatment). *t*-test on early vs late cells treated for 60 min, *p* = 0.3065. *t*-test on early vs late cells treated for 120 min, *p* = 0.8987. ns: not significant.

**Figure 3 biomolecules-09-00044-f003:**
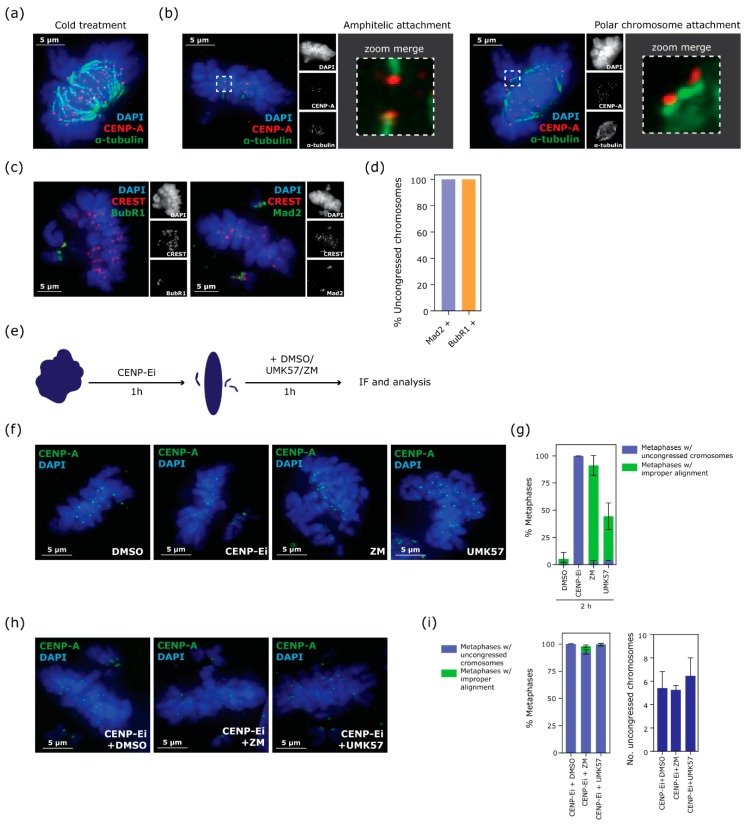
Perpetually uncongressed chromosomes are not syntelically attached. (**a**) Representative immunofluorescence image of prometaphase with uncongressed chromosomes after 10 min of cold treatment to remove unstable spindle microtubules. Cells are stained with CENP-A antibody to mark centromeres (red) and with α-tubulin antibody to mark spindle microtubules (green). (**b**) Representative immunofluorescence images of cells treated for 2 h with CENP-Ei and subsequently for 10 min with cold treatment. Zooms indicate kinetochore-microtubule attachments. Left panel highlights a correctly aligned chromosome which is amphitelically attached to the mitotic spindle. Right panel highlights a polar chromosome laterally attached to the mitotic spindle. (**c**) Representative images showing two major markers of spindle assembly checkpoint (SAC) activation in CENP-Ei treated cells (BubR1 and Mad2, green). (**d**) Percentage of uncongressed chromosomes that are Mad2 or BubR1 positive, one experiment (122 cells analysed). (**e**) Schematic of experimental procedure designed to investigate syntelic attachment of perpetually uncongressed chromosomes. (**f**) Representative immunofluorescence images of cells treated with CENP-Ei, MCAK potentiator (UMK57) and Aurora B inhibitor (ZM). (**g**) Percentage of erroneous metaphases obtained from single drug treatments. Blue bars indicate metaphases with uncongressed chromosomes, green bars indicate an improper alignment of chromosomes which results in abnormal metaphase shape. Mean and S.D. from three independent experiments (at least 150 metaphase cells analysed in total per each condition) are shown. (**h**) Representative immunofluorescence images of cells treated with combinations of drugs as indicated. (**i**) Percentage of erroneous metaphases and number of uncongressed chromosomes per prometaphase obtained from combination of drug treatments. Data show mean and S.D. from three independent experiments (at least 150 metaphase cells analysed in total per condition). IF: immunofluorescence.

**Figure 4 biomolecules-09-00044-f004:**
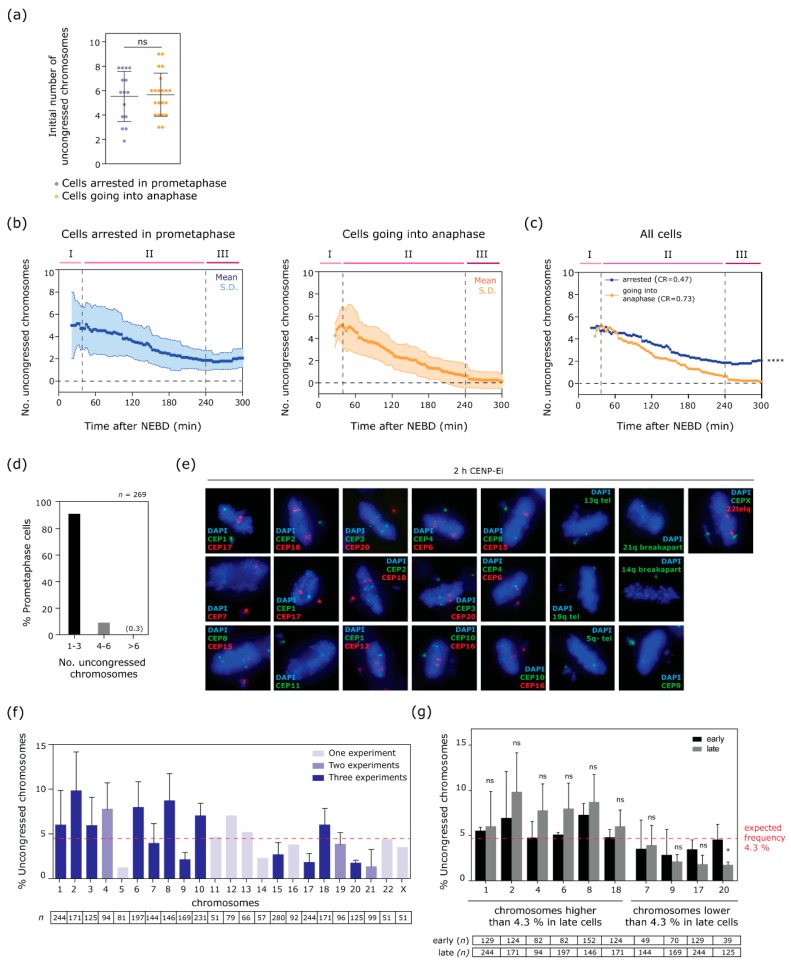
Larger chromosomes are more likely to remain perpetually uncongressed under CENP-E inhibition. (**a**) Initial number of uncongressed chromosomes in cells that remain arrested in prometaphase or undergo anaphase. Unpaired *t*-test, *p* = 1. (**b**) Comparison of uncongressed chromosome number during time in cells arrested in prometaphase (left graph) and in cells going into anaphase (right graph). (**c**) Overlay of the two populations of cells (mean) in (**b**). ****<0.0001, two-tailed Mann Whitney test. CR: chromosome congression rate in phase II. (**d**) Distribution of prometaphase cells with 1–3, 4–6 or more than 6 uncongressed chromosomes in fluorescence in situ hybridisation (FISH) experiments of cells treated with CENP-Ei for 2 h. (**e**) Example FISH images of all human chromosomes marked with centromere probes (CEP), subtelomere probes or breakapart probes. (**f**) Percentage of uncongressed chromosomes that are indicated. Dark violet bars represent 3 experiments, light violet bars represent 2 experiments, pale violet bars represent 1 experiment (at least 50 cells have been analysed for each experiment). Mean and S.D. are shown for data from 2 or 3 experiments, mean is shown for data from 1 experiment. A red dashed line indicates the expected rates of uncongression for each chromosome if each chromosome was equally affected (1/23 = 0.043). Bottom pane indicates the number of prometaphases analysed for each chromosome. Note that we adjusted chromosome-specific rates of congression for chromosome 12 to account for the observed trisomy rates for this chromosome. (**g**) Percentage of uncongressed chromosomes that are indicated in early prometaphase stage (4–6 chromosomes, black bars) or late prometaphase stage (1–3 chromosomes, grey bars); *t*-test for chromosome 1, *p* = 0.87; *t*-test for chromosome 2, *p* = 0.49; *t*-test for chromosome 4, *p* = 0.34; *t*-test for chromosome, 6 *p* = 0.27; *t*-test for chromosome, 8 *p* = 0.58; *t*-test for chromosome 18, *p* = 0.35; *t*-test for chromosome 7, *p* = 0.89; *t*-test for chromosome 17, *p* = 0.17; *t*-test for chromosome 20, *p* = 0.038. Mean and S.D. are shown for data from two or three independent experiments. Bottom pane indicates the number of prometaphases analysed for each chromosome in early (4–6 uncongressed chromosomes) and late (1–3 chromosomes) cells. Data from late cells are reproduced from (**f**) for comparison.

**Table 1 biomolecules-09-00044-t001:** Details of fluorescence in situ hybridisation (FISH) probes.

Chromosome	Probe	Chromosome Region	Probe Code
1	centromeric	1q12	LPE 001R/G
2	centromeric	2p11.1-q11.1	LPE 001R/G
3	centromeric	3p11.1-q11.1	LPE 001R/G
4	centromeric	4p11.1-q11.1	LPE 001R/G
5	telomeric	5qtel	LPT 05QG/R
6	centromeric	6p11.1-q11.1	LPE 001R/G
7	centromeric	7p11.1-q11.1	LPE 001R/G
8	centromeric	8p11.1-q11.1	LPE 001R/G
9	centromeric	9q12	LPE 001R/G
10	centromeric	10p11.1-q11.1	LPE 001R/G
11	centromeric	11p11.1-q11.1	LPE 001R/G
12	centromeric	12p11.1-q11.1	LPE 001R/G
13	telomeric	13qtel	LPT13QG/R
14	TCL1 breakapart	14q.32.13-q32.2	LPH 046-S
15	centromeric	15p11.1-q11.1	LPE 001R/G
16	centromeric	16p11.1-q11.1	LPE 001R/G
17	centromeric	17p11.1-q11.1	LPE 001R/G
18	centromeric	18p11.1-q11.1	LPE 001R/G
19	telomeric	19qtel	LPT 19QG/R
20	centromeric	20p11.1-q11.1	LPE 001R/G
21	AML breakapart	21q22.12	LPH 027-S
22	telomeric	22qtel	LPT 22QG/R
X	centromeric	Xp11.1-q11.1	LPE 001R/G

TCL1: T Cell Leukemia; AML: Acute Myeloid Leukemia.

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
