# Peer review of "Impaired CENP-E Function Renders Large Chromosomes More Vulnerable to Congression Failure"

_biomolecules, 2019, doi:10.3390/biom9020044_

Reviewer 1 Report

The manuscript by Tovini and McClelland investigates chromosome-specific dependence on CENP-E motor for congression in human cells. This is an interesting question since it is well established that only few chromosomes require CENP-E for congression. To my knowledge, this question has only been addressed so far in the simple placental mammal Indian muntjac, which has only 3 distinct chromosomes and thus individual chromosomes could be tracked in simple live cell experiments. Because human cells have man more chromosomes, many of which cannot be easily distinguished, here the authors used a combination of live and fixed-cell analysis with FISH for all human chromosomes. Intriguingly, the authors reported that human cells undergoing mitosis upon CENP-E inhibition experienced 4 distinct phases: a first rapid congression phase of most chromosomes, followed by a slow, but progressive phase in which ~1 chromosome/hour completed congression, until they eventually “arrest” with perpetual polar chromosome, ultimately culminating in cohesion fatigue. This analysis offers a nice characterization of CENP-E inhibition in human cells. The authors then investigate the reason for polar chomosomes to remain perpetually misaligned and found that this does not involve end-on attachments (of any type), is not due to centrosome shielding and is independent of kinetochore-microtubule dynamics. Interestingly, FISH analysis upon CENP-E inhibition revealed a bias towards large chromosomes to remain polar, with few exceptions. While the authors do not test any possible explanation for this bias, I think that this is an interesting study that would be of interest for publication in Biomolecules. More specific comments follow below.

Specific issues:

1-      While the authors show convincing data that centrosomes do not shield large chromosomes that remain perpetually polar, previous work in human cells (Barisic et al., NCB, 2014) has provided evidence from 4D-backtracking of polar chromosomes that the position of chromosomes at NEBD is a critical determinant. In this case, chromosomes that were more closely positioned to one of the poles and outside the interpolar axis tended to depend more on CENP-E for congression. This should be discussed or tested to explain the current bias. Moreover, the distinction between the position of uncongressed chromosomes relative to the poles (tested here) and the position of chromosomes relative to the poles at NEBD (tested by Barisic et al) should be clarified in the abstract and throughout the text to avoid confusion. As it reads, it looks like chromosome position relative to the poles is not important to determine CENP-E dependence.

2-      Introduction needs significant work. Some sentences are not well constructed and some concepts are simply inaccurate. Some examples: a) “kinetochores must interact with microtubules form opposite spindle poles to ensure proper alignment”, this is inaccurate, as shown by Kapoor et al., Science, 2006 and Cai et al., NCB, 2009 – hence the CENP-E pathway; b) “formation of kinetochore-microtubule binding”…this does not sound wright; c) “a bundle of microtubules emanates from each centrosome”, although a bundle of MTs does form, they do not emanate as bundles from the centrosome; d) “Once the interaction is made, kinesin-like proteins are recruited” - kinesin-like proteins are already at kinetochores prior to microtubule interaction; e) “end-on conversion to lateral attached”…this is quite controversial and the authors’ own data denies this model as even without CENP-E some polar chromosomes congress and align/bi-orient; f) the authors introduce drug treatments and then use as an example CENP-E heterozygous mice…

3-      Results 3.2. – the authors state that it is not clear why some chromosomes fail to align during the rapid congression phase. However, previous work (Barisic et al., NCB, 2014) has offered an explanation for this: peripheral chromosomes that cannot bi-orient soon after NEBD do not congress rapidly and rely on CENP-E for congression. This should be acknowledged. It would be interesting to test whether the alignment of a subset of polar chromosomes after CENP-E inhibition depends on chromokinesins.

4-      Figure 2c. misses statistical validation.

5-      Results 3.3. – it is reassuring that the authors never observed any end-on attachments at polar chromosomes, but this has been well established by prior works (two EM works by McEwen et al., MBoC, 2001; Putkey et al., Dev Cell, 2002 using antibody injections or CENP-E KO mouse fibroblasts; and one after CENP-E inhibition by Barisic et., NCB, 2014). In fact, Barisic et al. have shown that the lateral attachment of polar chromosomes actually depends on Dynein activity. These works should be cited in this context.

6-      Results 3.4. – the authors state that 56% of CENP-Ei cells manage to congress ALL chromosomes and proceed to anaphase. How sure can they be that ALL chromosomes aligned prior to anaphase? From my understanding, this conclusion was withdrawn based on the duration between NEBD-Ana onset. Previous works have shown that upon CENP-E inactivation/depletion, some cells do enter anaphase with few uncongressed chromosomes (Weaver et al., JCB, 2003; Maia et al., Chromosoma, 2010).

7-      Discussion – Yang et al., Curr Biol, 2007 should be cited together with ref. 26.

8-      Discussion – force balance between dynein and chromokinesins in CENP-Ei cells. The authors conclude that the balance must be tipped towards dynein. This has in fact been shown again by Barisic et al., NCB 2014 and should be acknowledged.

9-      Discussion – how do the authors envision that larger chromosomes might load more Dynein? Are there detectable differences in kinetochore structure/size on polar chromosomes? Because unattached KTs on polar chromosomes have expanded coronas, they might indeed recruit more Dynein, but this is simply a consequence of their unattached status rather than chromosome size. How does this model compare with the recent work in the Indian muntjac system (Drpic et al., Curr Biol, 2018) where it was shown that chromosomes with larger KTs depend less on CENP-E for congression? This should be discussed.

10-   Discussion – related to the previous point, the authors conclude that it appears that there is no correlation between centromere length and the propensity to remain polar (in human cells). However, we do not know what is actually the centromere length of the different human chromosomes, with the exception of the Y chromosome, which was only recently sequenced (Jain et al., Nat Biotech, 2018). Even the Y chromosome showed significant variability among different individuals and array size can vary up to 10 fold for the same chromosome in different individuals. Finally, we do not know how does centromere length correlates with kinetochore size in humans, with some works (e.g. Sullivan lab) suggesting that CENP-A deposition does not occur along the entire satellite array. For these reasons, this should be discussed with some caution not to withdraw premature conclusions that cannot be supported by data.

11-   Discussion – the chromosome territories model is interesting and in agreement with the previous finding by Barisic et al., relating chromosome positioning at NEBD and CENP-E dependence for congression. Maybe in RPE-1 cells large chromosome territories are indeed more peripheral and thus cannot bi-orient soon after NEBD, depending more on CENP-E.

Author Response

Rebuttal for ‘Impaired CENP-E Function Renders Large Chromosomes More Vulnerable to Congression Failure’

We kindly thank the reviewers for their thorough reading of our manuscript, insightful comments, and expedient review. We are happy that both reviewers find the study interesting and of significance for the readers of Biomolecules. We agree with all of the points raised by the reviewers and have amended the text and figures accordingly (see below for detailed responses, and amended text is also highlighted in yellow in the revised manuscript). Additionally we have performed a further analysis of bias against large chromosomes at early and late cells, to further characterise the likely cause of bias. We feel the manuscript is now much improved, and we hope the reviewers find our changes satisfactory.

Reviewer 1:

The manuscript by Tovini and McClelland investigates chromosome-specific dependence on CENP-E motor for congression in human cells. This is an interesting question since it is well established that only few chromosomes require CENP-E for congression. To my knowledge, this question has only been addressed so far in the simple placental mammal Indian muntjac, which has only 3 distinct chromosomes and thus individual chromosomes could be tracked in simple live cell experiments. Because human cells have man more chromosomes, many of which cannot be easily distinguished, here the authors used a combination of live and fixed-cell analysis with FISH for all human chromosomes. Intriguingly, the authors reported that human cells undergoing mitosis upon CENP-E inhibition experienced 4 distinct phases: a first rapid congression phase of most chromosomes, followed by a slow, but progressive phase in which ~1 chromosome/hour completed congression, until they eventually “arrest” with perpetual polar chromosome, ultimately culminating in cohesion fatigue. This analysis offers a nice characterization of CENP-E inhibition in human cells. The authors then investigate the reason for polar chomosomes to remain perpetually misaligned and found that this does not involve end-on attachments (of any type), is not due to centrosome shielding and is independent of kinetochore-microtubule dynamics. Interestingly, FISH analysis upon CENP-E inhibition revealed a bias towards large chromosomes to remain polar, with few exceptions. While the authors do not test any possible explanation for this bias, I think that this is an interesting study that would be of interest for publication in Biomolecules. More specific comments follow below.

Specific issues:

1-      While the authors show convincing data that centrosomes do not shield large chromosomes that remain perpetually polar, previous work in human cells (Barisic et al., NCB, 2014) has provided evidence from 4D-backtracking of polar chromosomes that the position of chromosomes at NEBD is a critical determinant. In this case, chromosomes that were more closely positioned to one of the poles and outside the interpolar axis tended to depend more on CENP-E for congression. This should be discussed or tested to explain the current bias. Moreover, the distinction between the position of uncongressed chromosomes relative to the poles (tested here) and the position of chromosomes relative to the poles at NEBD (tested by Barisic et al) should be clarified in the abstract and throughout the text to avoid confusion. As it reads, it looks like chromosome position relative to the poles is not important to determine CENP-E dependence.

We thank the reviewer for pointing out the lack of clarity regarding this point.

We have now amended the manuscript to clarify that position at NEBD is likely to be important in dictating the likelihood of being a polar chromosome initially, and discussed this in the context of Barisic et al’s findings:

Studies characterising the effects of impaired CENP-E function have shown efficient alignment for most chromosomes, but with a subset remaining close to the poles[5, 6], suggesting a differential dependency on CENP-E between chromosomes[7]. One cause of this variance is that chromosomes positioned close to the centrosome or outside the interpolar axis at nuclear envelope breakdown are particularly reliant on CENP-E function[5].

We have also clarified this in the abstract:

“It is known that inhibition of CENP-E notably impairs alignment for a subset of chromosomes, particularly those positioned close to the centrosome at nuclear envelope breakdown (‘polar chromosomes’) but it is not clear whether chromosome identity could influence this process.”……

“This bias is likely due to two contributing factors; an initial propensity of larger chromosomes to be peripheral and thus rely more upon CENP-E function to migrate to the metaphase plate, and additionally a bias between specific chromosomes’ ability to congress from a polar state.”

2-      Introduction needs significant work. Some sentences are not well constructed and some concepts are simply inaccurate. Some examples: a) “kinetochores must interact with microtubules form opposite spindle poles to ensure proper alignment”, this is inaccurate, as shown by Kapoor et al., Science, 2006 and Cai et al., NCB, 2009 – hence the CENP-E pathway; b) “formation of kinetochore-microtubule binding”…this does not sound wright; c) “a bundle of microtubules emanates from each centrosome”, although a bundle of MTs does form, they do not emanate as bundles from the centrosome; d) “Once the interaction is made, kinesin-like proteins are recruited” - kinesin-like proteins are already at kinetochores prior to microtubule interaction; e) “end-on conversion to lateral attached”…this is quite controversial and the authors’ own data denies this model as even without CENP-E some polar chromosomes congress and align/bi-orient; f) the authors introduce drug treatments and then use as an example CENP-E heterozygous mice…

We agree with these points and have amended the introduction accordingly:

During prometaphase, chromosomes congress to the spindle equator[1]. Although in mammalian cells chromosomes can congress before becoming bioriented[2], kinesin-like proteins precisely guide chromosome movement during the formation of the metaphase plate. Among them, Centromere-associated protein E (CENP-E) facilitates chromosome alignment by assisting their motion towards plus ends of microtubule bundles of the mitotic spindle[2, 3]. Moreover, CENP-E is involved, together with other kinesins, in promoting end-on conversion of chromosome-microtubule attachment [4].

Errors in the process of chromosome segregation can lead to aneuploidy, a key hallmark of cancer[10-12]. CENP-E heterozygous mice have been shown to display increased aneuploidy and tumour formation[13]. Moreover, common strategies to study aneuploidy in cellular models of cancer also employ alteration of CENP-E function.

3-      Results 3.2. – the authors state that it is not clear why some chromosomes fail to align during the rapid congression phase. However, previous work (Barisic et al., NCB, 2014) has offered an explanation for this: peripheral chromosomes that cannot bi-orient soon after NEBD do not congress rapidly and rely on CENP-E for congression. This should be acknowledged. It would be interesting to test whether the alignment of a subset of polar chromosomes after CENP-E inhibition depends on chromokinesins.

We agree that peripheral chromosomes are likely to depend more on CENP-E for congression and apologise for the lack of clarity and the omission of Barisic et al at this point. We have amended the text to clarify that we wondered why, of the initially peripheral chromosomes, some manage to congress while others do not:

Results:

The rapid formation of the pseudo-metaphase plate even when CENP-E function is impaired suggests that many chromosomes are able to rapidly bi-orient in the absence of lateral microtubule motion, potentially because they are already positioned near the equator of the cell at the moment of nuclear envelope breakdown. However some chromosomes fail to align during this rapid congression phase and remain at the poles (‘polar chromosomes’[6]). It is known that initial position at NEBD influences the propensity of chromosomes to fail initial congression and become polar[5]. However we were interested in understanding why a subset of these polar chromosomes were then able to congress - albeit at a reduced rate – during the progressive phase (phase II, Figure 1e), while others were left stranded at the centrosomes indefinitely (‘perpetually polar chromosomes’).

Discussion:

To date it has been shown that CENP-E inhibition impairs chromosome congression, with a characteristic subset of chromosomes that fail to initially congress, often as a result of being located outside the interpolar microtubules at NEBD [5]. Here, we used live cell imaging to show that of the initial polar chromosomes, there is a final set that fails to congress during prolonged prometaphase.

We thank the reviewer for the suggestion to test the contribution of chromokinesins to the bias. We agree this would be interesting and we are interested to test this in future studies to help elucidate the precise reasons for variance between chromosomes.

4-      Figure 2c. misses statistical validation.

This is now included in the Figure 2d legend.

Quantification of position of uncongressed chromosomes in early or late phase cells at 60 or 120 min of CENP-Ei treatment, two experiments (125 cells and 676 chromosomes analysed for 60 min treatment; 126 cells and 717 chromosomes analysed for 120 min treatment). T-test on early vs late cells treated for 60 min p = 0.3065. T-test on early vs late cells treated for 120 min p = 0.8987.

5-      Results 3.3. – it is reassuring that the authors never observed any end-on attachments at polar chromosomes, but this has been well established by prior works (two EM works by McEwen et al., MBoC, 2001; Putkey et al., Dev Cell, 2002 using antibody injections or CENP-E KO mouse fibroblasts; and one after CENP-E inhibition by Barisic et., NCB, 2014). In fact, Barisic et al. have shown that the lateral attachment of polar chromosomes actually depends on Dynein activity. These works should be cited in this context.

We apologise for the omission and have included reference to these works in the revised text:

CENP-E inhibition is proposed to impair the conversion from lateral to end-on kinetochore-microtubule attachment[4] and therefore uncongressed chromosomes are known to be laterally attached[5, 19, 20].

6-      Results 3.4. – the authors state that 56% of CENP-Ei cells manage to congress ALL chromosomes and proceed to anaphase. How sure can they be that ALL chromosomes aligned prior to anaphase? From my understanding, this conclusion was withdrawn based on the duration between NEBD-Ana onset. Previous works have shown that upon CENP-E inactivation/depletion, some cells do enter anaphase with few uncongressed chromosomes (Weaver et al., JCB, 2003; Maia et al., Chromosoma, 2010).

We agree this was not clear in the original manuscript. In fact, we judged whether anaphase onset occurred in the presence, or absence of uncongressed chromosomes from live cell imaging. Further, since we always saw strong localisation of Mad2 and BubRI at uncongressed chromosomes (Figure 3) we suspect the SAC is active until congression is complete, at least in RPE1 cells and under our conditions (20 nM CENP-Ei). However it is true that within the time resolution of our movies (3 min timelapse) we cannot rule out that some cells do enter anaphase with uncongressed chromosomes. We have amended the text to clarify these points:

However, live cell analyses showed that a proportion (56%) of CENP-Ei-treated cells did proceed to anaphase during this time (see Figure 1d). In accordance with the efficient loading of mitotic checkpoint proteins Mad2 and BubR1 at polar chromosomes (see Figure 3c,d), we noted from live cell imaging that cells correctly aligned all chromosomes before undergoing anaphase (see Figure 1) at least within the time resolution of the time-lapse filming (3 minutes).

7-      Discussion – Yang et al., Curr Biol, 2007 should be cited together with ref. 26.

This is now cited and we thank the reviewer for the suggestion.

8-      Discussion – force balance between dynein and chromokinesins in CENP-Ei cells. The authors conclude that the balance must be tipped towards dynein. This has in fact been shown again by Barisic et al., NCB 2014 and should be acknowledged.

This is now cited and we thank the reviewer for pointing out our omission:

Interestingly, although large chromosomes should potentially benefit from a higher polar ejection force, the observed enrichment of these chromosomes at centrosomes would support previous observation that the force balance is in fact tipped towards dynein winning[5].

9-      Discussion – how do the authors envision that larger chromosomes might load more Dynein? Are there detectable differences in kinetochore structure/size on polar chromosomes?  Because unattached KTs on polar chromosomes have expanded coronas, they might indeed recruit more Dynein, but this is simply a consequence of their unattached status rather than chromosome size. How does this model compare with the recent work in the Indian muntjac system (Drpic et al., Curr Biol, 2018) where it was shown that chromosomes with larger KTs depend less on CENP-E for congression? This should be discussed.

At present it is not clear how or whether larger chromosomes load more dynein. As the reviewer points out, we unfortunately cannot simply analyse kinetochore composition of polar versus aligned chromosomes due to the gross changes in structure (kinetochore expansion) present on these uncongressed chromosomes. It would be interesting in future studies to perform immunofluorescence coupled to FISH of specific centromeres to quantify chromosome-specific kinetochore composition.

Regarding the Drpic et al recent study, we agree that our results should be discussed in regards to this finding specifically and have amended the discussion text:

Large chromosomes may inherently load more dynein to compensate for their size. Under normal conditions this is counterbalanced by CENP-E function, but following CENP-E inhibition large chromosomes are vulnerable to congression failure. Interestingly, recent work showed that chromosomes with larger centromeres depend less on CENP-E for congression in the Indian Muntjak system[9]. This would suggest that even if large chromosomes do load more dynein, that this might not translate directly into a ‘larger’ kinetochore per se (in terms of providing additional MT binding sites).

10-   Discussion – related to the previous point, the authors conclude that it appears that there is no correlation between centromere length and the propensity to remain polar (in human cells). However, we do not know what is actually the centromere length of the different human chromosomes, with the exception of the Y chromosome, which was only recently sequenced (Jain et al., Nat Biotech, 2018). Even the Y chromosome showed significant variability among different individuals and array size can vary up to 10 fold for the same chromosome in different individuals. Finally, we do not know how does centromere length correlates with kinetochore size in humans, with some works (e.g. Sullivan lab) suggesting that CENP-A deposition does not occur along the entire satellite array. For these reasons, this should be discussed with some caution not to withdraw premature conclusions that cannot be supported by data.

We agree entirely with these points about ‘centromere length’ and ‘size’. We have amended the text accordingly:

Therefore there does not appear to be a clear correlation between centromere length as estimated from centromeric array lengths and propensity to remain polar. However it is not currently possible to make robust conclusions regarding contribution to congression bias from differences in centromere length or size, since these parameters are not currently well defined. Future studies will be required to functionally test centromeric differences that could contribute to variance in congression ability.

11-   Discussion – the chromosome territories model is interesting and in agreement with the previous finding by Barisic et al., relating chromosome positioning at NEBD and CENP-E dependence for congression. Maybe in RPE-1 cells large chromosome territories are indeed more peripheral and thus cannot bi-orient soon after NEBD, depending more on CENP-E.

We agree with this interpretation, and we have now cited the Barisic reference when discussing this point. We also now explicitly state this hypothesis which we believe was not clearly stated originally:

It is therefore a strong possibility that the size bias may reflect the initial arrangement of chromosomes in the nucleus; small chromosomes may be more likely to be positioned close to the site of the metaphase plate and be more likely to encounter MTs at the correct geometry to form end-on attachments even in the absence of CENP-Ei. In contrast, large chromosomes positioned at the periphery of the nucleus would be more likely to become polar since the initial position relative to the centrosome is important in dictating the dependency upon CENP-E for congression[9].

Reviewer 2 Report

The take-home message from this work is that the probability of a chromosome to become perpetually monooriented in cells treated with a rigor-state inhibitor of the kinesin CenpE is higher for larger chromosomes.  A corollary of this difference is that chromosomal inhibition of CenpE leads to a biased chromosome mis-segregation. This is an important conclusion and it offers a new perspective on the origin of non-random chromosome distribution in certain types of cells or under various experimental conditions. Therefore, the work reported in the manuscript is highly significant and it will be appreciated by the scholars of cell division and cancer researchers alike. However, several important issues need to be addressed prior to publication. Most of these issues can be resolved by textual edits, other points are more substantial. To this reviewer, the main issue is the apparent numeric inconsistencies among the data presented in different figures and the lack of clear evidence that the analyses were focused on those chromosomes that are perpetually monooriented.   

1. Dynamics of mitotic progression and congression should be better illustrated in Fig.1. One current deficiency is that the plot depicting the cumulative probability function of mitotic exit ends abruptly at 240 min. This plot should be extended to the full range 480 min. As evident from Fig.1e, the authors possess a large number of live-cell recordings during 240-480 min so the data are readily available. Revealing the shape of CDF is very important in the context of this manuscript. For example, it will help to better assess whether the cells really enter a different physiological state after the initial 200 min. The authors state that they “rarely observed any congression events after 200 min“, what about mitotic exit?  The presented part of mitotic exit PDF appears to be linear, which is peculiar. Does it bend sharply at 240 min (indicates a binary switch) or is the shape consistent with normal distribution? Providing this information is important because it relates to the curve shown in Fig.1e and the main issue in the manuscript is the apparent inconsistency between this curve and the data in the rest of the figures.

2. There is a discrepancy between Fig.1e and Fig.2 that needs to be explained or resolved. According to Fig.1e, cells that contain 1-3 chromosomes after 60-min incubation in CenpE inhibitor are not typical. This range of polar chromosomes numbers is in the second standard deviation from the mean. Even at 120 min such a low number of polar chromosomes would be unusual for a cell that is destined to remain perpetually arrested. Therefore, it appears that the distribution of inner vs. outer polar chromosomes is determined in those cells that congress their chromosomes at a higher than typical rate. A natural expectation is that these are the cells that will exit mitosis, not those that are perpetually arrested. If so, interpretation of the data would be affected by the biased analysis.

3. Fig.4a-b also leads to some confusion that should be resolved. The plots show that the initial number of chromosomes and the rate of congression during Phase II (200 min duration) are identical in cells that exist mitosis vs. those that arrest perpetually. Thus, cells must exit mitosis in the presence of polar chromosomes. However, the authors clearly state that cells exit only after all chromosomes have congressed (p.9, lines 248-250). If the ‘exiting’ cells manage to congress all chromosomes in less than 240 min (56% of cells exit) while ~2 chromosomes remain after 240 min in the ‘arrested’ cells, then either the initial number of polar chromosomes or the rate of congression must be different in these classes. The data presented in Fig. 1 vs. Fig. 4 appear to be mutually inconsistent.

4. Related to point 2, FISH analyses were conducted in cells with a low number of polar chromosomes after a relatively short treatment (120 min) with CenpE inhibitor. According to Fig.1c,e, these cells are progressing through spindle assembly faster than most and therefore there is a good chance that they are destined to exit. If so, the analysis might be biased towards chromosomes that are not perpetually polar. This should be considered in the interpretation of the results. The conclusion favored by the authors (that large chromosomes cannot congress upon CenpE inhibition) requires that the analysis focuses on perpetually polar chromosomes. Enrichment of larger chromosomes in the polar population is actually consistent with the idea that CenpE is more important for the ‘peripheral’ chromosomes that are excluded from the central parts of the spindle by the ejection force (Maiato’s work). Spindle ejection force is expected to affect larger chromosomes to a larger degree. In the context of the current manuscript it is important to prove that larger chromosomes are specifically enriched among those that become perpetually monooriented not just in the population of polar chromosomes.

An unusual yet highly significant problem:  The authors state that hTERT-RPE1 cells were a gift from Dr. S. Godinho (line 68). This statement is a problem because the original purchaser of the cell line (presumably Dr. Godinho) was required to sign a very restrictive license agreement that specifically prohibits sharing these cells even with other labs at the same institution. These restrictions are due to the patent that regulates the use of hTERT. ATCC aggressively pursues unlicensed use of hTERT cell lines. Indeed, this Reviewer has experienced the consequences of not following the rules when it comes to hTERT cell lines. We shared hTERT-RPE1 cells with a collaborator who later used this cell line in a separate study. When ATCC accidentally discovered that the cells were used by an unlicensed laboratory they halted all transactions of my collaborator’s institution (a major university) until my collaborator acquired her own hTERT license at full price. Fortunately, my name was not mentioned in the publication and my collaborator refused to reveal to ATCC where she acquired the cells. Otherwise, as ATCC informed my collaborator, my license would be annulled, and I’d be required to re-acquire it at full price. Germaine here is that the statement jeopardizes both the authors who used hTERT-RPE1 without a license and Susanna who transferred the cell line to a third party in clear violation of her purchasing agreement. It is just a matter of time before ATCC discovers the unlicensed use.

Less important points

Line 72. Passages 10-25 for hTERT-RPE1 do not exist. The earliest commercially available passage back in 2001 was 118.5 (used to be available through Clontech prior to the transfer of the stock to ATCC). The authors must specify what they consider to be passage #1.

Opening sentences of the Intro (lines 31-33) are not justified.  Formation of the equatorial plate has never been shown to be a “key step” in the “ability to maintain chromosome segregation fidelity”. Indeed, a perfect metaphase plate assembles in cells with numerous merotelic attachments, which is the most common cause of chromosome mis-segregation.

The statement that “a bundle of microtubules … randomly probes the presence of a free kinetochore” (lines 35-36) is factually incorrect. Astral microtubules are never bundled while bundles of microtubules never exhibit the pattern of dynamic instability required for the search. Also, please notice the typo: “Multiple mechanisms has been proposed…”

“Once the interaction is made, kinesin-like proteins are recruited…” (line38). The statement is factually incorrect and conceptually flawed. Actually, concentrations of CenpE and dynein at the kinetochore decrease upon attachment. If motors were recruited only after the initial contact with microtubules, then how would the initial contact be made and maintained while the motors are being recruited?

Lines 39-41. CenpE is not just “assisting [chromosome] motion along microtubules”, it drives this movement. Further, the statement that CenpE is “required for end-on conversion to lateral attached chromosomes” directly contradicts the very next sentence. If CenpE is required (i.e., is essential) for the lateral to end-on conversion, then how could cells with impaired CenpE achieve “efficient alignment for most chromosomes”? The authors should be careful in choosing specific terms: CenpE facilitates (promotes, assists, etc.) formation of end-on attachments but it clearly is not required for this process. In contrast, gliding of kinetochores alongside microtubule bundles towards the plus ends requires CenpE (it drives this type of movement). 

Lines 49-52. These sentences seem to be disconnected. This reviewer fails to see why chromosomal instability observed in CenpE heterozygous mice is an “example” of “common strategies to study chromosome segregation errors” that “involve drug treatments to induce losses of chromosomes”.  The transition makes no sense.

Lines 156-157 “in the absence of lateral microtubule motion”. As formulated, the statement is incorrect. Inhibition of CenpE blocks only some types of lateral gliding. Dynein is still active, and kinetochores do glide towards the minus ends in CenpE-inhibited cells. 

Lines 194-195. “are likely to be laterally attached”. If lateral interactions are a type of ‘attachment’ then the presence of Mad2 and BubR1 is not “in agreement with an unattached status” (line 207). This actually illustrates a larger issue created by less than strict adherence to proper terminology. In places consecutive sentences become illogical: “… in agreement with an unattached status (Figure 3c,d). These 207 analyses suggested that polar chromosomes were likely to be laterally attached”. This literally states that UNATTACHED kinetochores are ATTACHED.

Lines 275-276. Description of the effect on chr.18 is repeated twice in consecutive sentences.

The meaning of times marked as “a” and “b” in Fig.4 b is not explained.

Author Response

Rebuttal for ‘Impaired CENP-E Function Renders Large Chromosomes More Vulnerable to Congression Failure’

We kindly thank the reviewers for their thorough reading of our manuscript, insightful comments, and expedient review. We are happy that both reviewers find the study interesting and of significance for the readers of Biomolecules. We agree with all of the points raised by the reviewers and have amended the text and figures accordingly (see below for detailed responses, and amended text is also highlighted in yellow in the revised manuscript). Additionally we have performed a further analysis of bias against large chromosomes at early and late cells, to further characterise the likely cause of bias. We feel the manuscript is now much improved, and we hope the reviewers find our changes satisfactory.

Reviewer 2:

The take-home message from this work is that the probability of a chromosome to become perpetually monooriented in cells treated with a rigor-state inhibitor of the kinesin CenpE is higher for larger chromosomes.  A corollary of this difference is that chromosomal inhibition of CenpE leads to a biased chromosome mis-segregation. This is an important conclusion and it offers a new perspective on the origin of non-random chromosome distribution in certain types of cells or under various experimental conditions. Therefore, the work reported in the manuscript is highly significant and it will be appreciated by the scholars of cell division and cancer researchers alike. However, several important issues need to be addressed prior to publication. Most of these issues can be resolved by textual edits, other points are more substantial. To this reviewer, the main issue is the apparent numeric inconsistencies among the data presented in different figures and the lack of clear evidence that the analyses were focused on those chromosomes that are perpetually monooriented.   

1. Dynamics of mitotic progression and congression should be better illustrated in Fig.1. One current deficiency is that the plot depicting the cumulative probability function of mitotic exit ends abruptly at 240 min. This plot should be extended to the full range 480 min. As evident from Fig.1e, the authors possess a large number of live-cell recordings during 240-480 min so the data are readily available. Revealing the shape of CDF is very important in the context of this manuscript. For example, it will help to better assess whether the cells really enter a different physiological state after the initial 200 min. The authors state that they “rarely observed any congression events after 200 min“, what about mitotic exit?  The presented part of mitotic exit PDF appears to be linear, which is peculiar. Does it bend sharply at 240 min (indicates a binary switch) or is the shape consistent with normal distribution? Providing this information is important because it relates to the curve shown in Fig.1e and the main issue in the manuscript is the apparent inconsistency between this curve and the data in the rest of the figures.

We agree that the kinetics of congression and anaphase entry should be displayed more clearly. We also apologise that in the original version of Figure 1d we inadvertently displayed only those cells that entered anaphase rather than the whole cell population. This has now been amended. Although we do possess movies from some cells for the full 480 minute duration of the movie, the majority of prometaphase cells were tracked for less time than this (due to later nuclear envelope breakdown time, or cells moving off-screen). To clarify this, we now display only data for cells we could track for at least 300 minutes following NEBD, or that entered anaphase within this time (20 of 21 cells that ultimately enter anaphase do so within 300 minutes) in Figure 1d-h. Those cells that fail to enter anaphase by 300 minutes all (with the exception of one cell that enters at t=438 minutes) remain perpetually arrested in prometaphase with uncongressed chromosomes. A subset of these arrested cells that we can follow past 300 minutes begin to display cohesion fatigue, that further prevents the alignment of all chromosomes and likely terminally activates the SAC (see new Figure S1). Therefore it is very unlikely that any cells will enter anaphase after this time, although we assume that after a further several hours we would begin to see mitotic exit in the form of failed anaphase. We now include Supplementary Figure 1 that shows the individual cell congression kinetics to further illustrate these cell behaviours.

2. There is a discrepancy between Fig.1e and Fig.2 that needs to be explained or resolved. According to Fig.1e, cells that contain 1-3 chromosomes after 60-min incubation in CenpE inhibitor are not typical. This range of polar chromosomes numbers is in the second standard deviation from the mean. Even at 120 min such a low number of polar chromosomes would be unusual for a cell that is destined to remain perpetually arrested. Therefore, it appears that the distribution of inner vs. outer polar chromosomes is determined in those cells that congress their chromosomes at a higher than typical rate. A natural expectation is that these are the cells that will exit mitosis, not those that are perpetually arrested. If so, interpretation of the data would be affected by the biased analysis.

We agree that directly extrapolating from live cell data would predict only a small fraction of cells with 1-3 uncongressed chromosomes at 120 minutes. However in fact we see a good proportion of cells with 1-3 uncongressed chromosome numbers in fixed cell analyses at this timepoint, (now presented as an additional graph in Figure 2c). This difference between live and fixed cell analysis is likely because some cells are already in prometaphase when CENP-Ei is added in the fixed cell experiments. However we agree that it is possible that the group of cells we class as ‘late’ could be more likely to be those that are destined for anaphase onset, which could skew the interpretation. We think this is unlikely to be a major factor however, since cells destined to enter anaphase start with the same number of uncongressed chromosomes, and possess a similar rate of congression when compared with cells that will arrest (now Figure 4a,b). Furthermore, in this particular analysis, the majority of chromosomes are not shielded in early or late cells at either timepoint, suggesting that any potential bias in our analysis would be unlikely to alter the major conclusion of this experiment. (Please also see point 4 below for a related discussion).

3. Fig.4a-b also leads to some confusion that should be resolved. The plots show that the initial number of chromosomes and the rate of congression during Phase II (200 min duration) are identical in cells that exist mitosis vs. those that arrest perpetually. Thus, cells must exit mitosis in the presence of polar chromosomes. However, the authors clearly state that cells exit only after all chromosomes have congressed (p.9, lines 248-250). If the ‘exiting’ cells manage to congress all chromosomes in less than 240 min (56% of cells exit) while ~2 chromosomes remain after 240 min in the ‘arrested’ cells, then either the initial number of polar chromosomes or the rate of congression must be different in these classes. The data presented in Fig. 1 vs. Fig. 4 appear to be mutually inconsistent.

We apologise for the lack of clarity in our original presentation of these data.

To clarify this point we now present extended congression graphs for 300 minutes-post NEBD, (limited to 300 minutes for the same reasons outlined above in point 1), which encompasses almost the entire period of time during which cells pass into anaphase (20 of 21 anaphase cells enter anaphase within 300 minutes). Originally we presented  data only until 200 minutes post-NEBD, since after this time cells that began to enter anaphase (and were initially removed from the analysis at this point) caused the average chromosome numbers to fluctuate upwards, rendering it difficult to interpret the kinetics after this point. We realise that this original presentation inadvertently gave the impression that cells that enter anaphase never reach an uncongressed chromosome number of zero. To correct this we now maintain cells that enter anaphase in the congression rate analysis, assigning them an uncongressed chromosome number of zero (revised Figure 4b, ‘cells going into anaphase’). This now clearly shows that these cells congress all chromosomes, in agreement with our data from live cell imaging showing that cells do not enter anaphase with uncongressed chromosomes (at least within the 3 minute time resolution), and that all unaligned chromosomes robustly mount an active mitotic checkpoint (Figure 3c,d).

Regarding the rate of congression – this was previously inadvertently calculated without taking into account cells that reached zero uncongressed chromosomes (see above). We have now recalculated these rates and now show that they are in fact subtly different (which we would expect given that anaphase destined cells contain polar chromosomes that are more likely to congress), and present this amended data in Figure 4 and the text:

There was no difference in the number of uncongressed chromosomes at the start of the progressive congression phase (Figure 4a). Both classes of cells also proceeded in a similarly progressive and consistent manner through congression phase (Phase II), though we noted a small difference in the rate of congression (Figure 4b,c). These similarities suggested that the factor influencing whether cells could congress all their chromosomes and proceed to anaphase was not a systemic state related to the time cells had spent in prometaphase, and was not a function of the initial number of polar chromosomes.”

4. Related to point 2, FISH analyses were conducted in cells with a low number of polar chromosomes after a relatively short treatment (120 min) with CenpE inhibitor. According to Fig.1c,e, these cells are progressing through spindle assembly faster than most and therefore there is a good chance that they are destined to exit. If so, the analysis might be biased towards chromosomes that are not perpetually polar. This should be considered in the interpretation of the results. The conclusion favored by the authors (that large chromosomes cannot congress upon CenpE inhibition) requires that the analysis focuses on perpetually polar chromosomes. Enrichment of larger chromosomes in the polar population is actually consistent with the idea that CenpE is more important for the ‘peripheral’ chromosomes that are excluded from the central parts of the spindle by the ejection force (Maiato’s work). Spindle ejection force is expected to affect larger chromosomes to a larger degree. In the context of the current manuscript it is important to prove that larger chromosomes are specifically enriched among those that become perpetually monooriented not just in the population of polar chromosomes.

We thank the reviewer for these insightful points.

As mentioned above in response to point 2, live imaging predicts only a small fraction of cells with 1-3 uncongressed chromosomes at 120 minutes. As above however, we see a good proportion of cells with 1-3 uncongressed chromosome numbers in FISH analyses at this timepoint (now presented as an additional graph in Figure 4d). We also note there are some differences in uncongressed number distribution between the IF (Figure 2c) and FISH (Figure 4d) experiments which may result from the different substrate (glass slides versus coverslips). There may also be slight underestimation of uncongressed chromosome number in the FISH analyses since we do not have an all centromere marker in these experiments. Ultimately though cells with 1-3 uncongressed chromosomes in fixed cell analyses do not represent an unusual minority, therefore we do not feel there is bias in this regard.

Nevertheless we agree that cells with 1-3 uncongressed chromosomes are likely a mix of cells destined to exit, and those destined to arrest. Thus the bias towards larger chromosomes remaining uncongressed in Figure 4f likely reflects a combination of two distinct variables; (i) the propensity for a given chromosome to be polar initially, and (ii) the ability of specific chromosomes to congress from a polar starting position.

We feel that there is likely to be contribution from both these aspects; First, there is a likely connection between chromosome size and propensity to initially form a polar chromosome position due to nuclear territory theory (see discussion). Second, the fact that cells that do, or do not, ultimately congress all chromosomes start with an identical average number of polar chromosomes (now Figure 4a) suggests a bias in the ability of some of these chromosome to congress. By contrast, if all chromosomes were equally able to congress, we would have expected a correlation between a low number of polar chromosomes, and the ability to enter anaphase. We have now amended the text to clarify that the bias could be contributed by either or both of these variables:

Thus the bias towards larger chromosomes remaining uncongressed in likely reflects a combination of two distinct variables; (i) the propensity for a given chromosome to be polar initially, and (ii) the ability of specific chromosomes to congress from a polar starting position.

As the reviewer points out there is also a likelihood that larger chromosomes may be peripheral due to being more greatly affected by the spindle ejection force. We thank the reviewer for this point and now include it in the discussion:

This suggests that the position of chromosomes in the nucleus could be relevant for their increased dependency on CENP-E, but that other factors may also be involved. Chromosomes can be excluded from central parts of the spindle by arm ejection forces[5] which would be expected to affect large chromosomes more, therefore this could be another contributing factor to the propensity of large chromosomes to become polar at NEBD.

We agree that it would be ideal to discover the identity of perpetually polar chromosomes specifically. Future studies using live cell markers of specific chromosomes will be important in fully exploring the identity of perpetually polar chromosomes, however for the present study we are limited to fixed cell analyses to identify specific chromosomes.

However, as a step further toward this we have now performed an additional analysis with our existing 120 minute CENP-Ei FISH slides to test the contribution of the two variables to bias more directly. This is now presented in Figure 4g and the results section:

Cells with 1-3 uncongressed chromosomes analysed in these FISH experiments are likely a mix of cells destined to enter anaphase, and those destined to arrest with perpetually polar chromosomes. Thus the bias towards larger chromosomes remaining uncongressed likely reflects a combination of two distinct variables; (i) the propensity for a given chromosome to be polar initially, and (ii) the ability of specific chromosomes to congress from a polar starting position. To test the contribution of these variables we analysed the uncongression bias in cells with 4-6 uncongressed chromosomes (‘early’ cells) and compared this to data collected from cells with 1-3 uncongressed chromosomes (‘late’ cells (Figure 4f)). We specifically asked; for chromosomes that are uncongressed above a rate of 4.3% in late cells (e.g. 1, 2, 4, 6, 8 and 18) what is the uncongression bias in ‘early’ versus ‘late’ cells? Interestingly we see a clear general trend (although most comparisons do not reach statistical significance); bias is lower in early cells, suggesting that initial positioning is not the only factor, and that these chromosomes are additionally less able to congress from a polar position. Taken together these data suggest that large chromosomes appear to be more likely to (i) become polar initially after NEBD and also (ii) fail to congress from a polar position.

Discussion:

Analysis of chromosome identity revealed a bias towards larger chromosomes becoming arrested at the centrosome, which appears to be contributed to by both an initial propensity to become polar, coupled to a reduction in congression efficiency from a polar state (Figure 4g).

An unusual yet highly significant problem:  The authors state that hTERT-RPE1 cells were a gift from Dr. S. Godinho (line 68). This statement is a problem because the original purchaser of the cell line (presumably Dr. Godinho) was required to sign a very restrictive license agreement that specifically prohibits sharing these cells even with other labs at the same institution. These restrictions are due to the patent that regulates the use of hTERT. ATCC aggressively pursues unlicensed use of hTERT cell lines. Indeed, this Reviewer has experienced the consequences of not following the rules when it comes to hTERT cell lines. We shared hTERT-RPE1 cells with a collaborator who later used this cell line in a separate study. When ATCC accidentally discovered that the cells were used by an unlicensed laboratory they halted all transactions of my collaborator’s institution (a major university) until my collaborator acquired her own hTERT license at full price. Fortunately, my name was not mentioned in the publication and my collaborator refused to reveal to ATCC where she acquired the cells. Otherwise, as ATCC informed my collaborator, my license would be annulled, and I’d be required to re-acquire it at full price. Germaine here is that the statement jeopardizes both the authors who used hTERT-RPE1 without a license and Susanna who transferred the cell line to a third party in clear violation of her purchasing agreement. It is just a matter of time before ATCC discovers the unlicensed use.

We appreciate the reviewer pointing out this important issue. In fact we have already purchased RPE-1 from ATCC. Funnily enough this line was incredibly unreliable in our hands, rapidly developing aneuploidy and displaying unusual mitotic behaviour. We contacted ATCC about this but they refuse to give a refund as they only state this line is ‘roughly’ diploid. Therefore for practical reasons we used the cells from our colleague. Nevertheless we have purchased the license and can state we purchased cells from ATCC. We cannot think of a better way around this and we will certainly bear this in mind for the future.

Less important points

Line 72. Passages 10-25 for hTERT-RPE1 do not exist. The earliest commercially available passage back in 2001 was 118.5 (used to be available through Clontech prior to the transfer of the stock to ATCC). The authors must specify what they consider to be passage #1.

We have removed this statement regarding passage number. Our original statement was simply referring to passages since our lab obtained the cells.

Opening sentences of the Intro (lines 31-33) are not justified.  Formation of the equatorial plate has never been shown to be a “key step” in the “ability to maintain chromosome segregation fidelity”. Indeed, a perfect metaphase plate assembles in cells with numerous merotelic attachments, which is the most common cause of chromosome mis-segregation.

The statement that “a bundle of microtubules … randomly probes the presence of a free kinetochore” (lines 35-36) is factually incorrect. Astral microtubules are never bundled while bundles of microtubules never exhibit the pattern of dynamic instability required for the search. Also, please notice the typo: “Multiple mechanisms has been proposed…”

“Once the interaction is made, kinesin-like proteins are recruited…” (line38). The statement is factually incorrect and conceptually flawed. Actually, concentrations of CenpE and dynein at the kinetochore decrease upon attachment. If motors were recruited only after the initial contact with microtubules, then how would the initial contact be made and maintained while the motors are being recruited?

Lines 39-41. CenpE is not just “assisting [chromosome] motion along microtubules”, it drives this movement. Further, the statement that CenpE is “required for end-on conversion to lateral attached chromosomes” directly contradicts the very next sentence. If CenpE is required (i.e., is essential) for the lateral to end-on conversion, then how could cells with impaired CenpE achieve “efficient alignment for most chromosomes”? The authors should be careful in choosing specific terms: CenpE facilitates (promotes, assists, etc.) formation of end-on attachments but it clearly is not required for this process. In contrast, gliding of kinetochores alongside microtubule bundles towards the plus ends requires CenpE (it drives this type of movement). 

Lines 49-52. These sentences seem to be disconnected. This reviewer fails to see why chromosomal instability observed in CenpE heterozygous mice is an “example” of “common strategies to study chromosome segregation errors” that “involve drug treatments to induce losses of chromosomes”.  The transition makes no sense.

Lines 156-157 “in the absence of lateral microtubule motion”. As formulated, the statement is incorrect. Inhibition of CenpE blocks only some types of lateral gliding. Dynein is still active, and kinetochores do glide towards the minus ends in CenpE-inhibited cells.

We agree with these points and have amended the introduction accordingly:

The ability to maintain chromosome segregation fidelity is a major feature of mitosis. During prometaphase, chromosomes congress towards the spindle equator[1].

Although in mammalian cells chromosomes can congress before becoming bioriented[2], kinesin-like proteins precisely guide chromosome movement during the formation of the metaphase plate. Among them, Centromere-associated protein E (CENP-E) facilitates chromosome alignment by assisting their motion towards plus ends of microtubule bundles of the mitotic spindle[2, 3]. Moreover, CENP-E is involved, together with other kinesins, in promoting end-on conversion of chromosome-microtubule attachment [4].

Errors in the process of chromosome segregation can lead to aneuploidy, a key hallmark of cancer[10-12]. CENP-E heterozygous mice have been shown to display increased aneuploidy and tumour formation[13]. Moreover, common strategies to study aneuploidy in cellular models of cancer also employ alteration of CENP-E function.

Lines 194-195. “are likely to be laterally attached”. If lateral interactions are a type of ‘attachment’ then the presence of Mad2 and BubR1 is not “in agreement with an unattached status” (line 207). This actually illustrates a larger issue created by less than strict adherence to proper terminology. In places consecutive sentences become illogical: “… in agreement with an unattached status (Figure 3c,d). These 207 analyses suggested that polar chromosomes were likely to be laterally attached”. This literally states that UNATTACHED kinetochores are ATTACHED.

We agree that there is a lack of strict adherence to terminology regarding lateral associations with MTs. To avoid the illogical nature of this sentence we have referred to an ‘unattached status’ as a lack of end-on attachment:

Next, we examined the mitotic checkpoint status of polar chromosomes using antibodies to Mad2 and BubR1 which revealed that the vast majority of KTs efficiently loaded Mad2 and BubR1 in agreement with a lack of end-on MT attachment (Figure 3c,d).

Lines 275-276. Description of the effect on chr.18 is repeated twice in consecutive sentences.

We have now fixed this repetition

The meaning of times marked as “a” and “b” in Fig.4 b is not explained.

We apologise for this omission, and we have now clearly marked the times bounded by these phases in Figure 4b,c.